# Understanding Entertainment Trends during COVID-19 in Saudi Arabia

**Amaal Aldawod, Raseel Alsakran and Hend Alrasheed ***

Department of Information Technology, King Saud University, Riyadh 11362, Saudi Arabia;
442203073@student.ksu.edu.sa (A.A.); 439203964@student.ksu.edu.sa (R.A.)
* Correspondence: halrasheed@ksu.edu.sa

**Abstract:** Studying people's opinions is a growing research field. This area of research is known as sentiment analysis. The COVID-19 pandemic changed everything around the world and reduced social contact among people. Citizens and residents of Saudi Arabia experienced high stress during the pandemic, seeking entertainment via games and publishing their activities on social media platforms such as Twitter. In this paper, we focus on applying the Mazajak sentiment analyzer on tweets containing game keywords in Arabic collected using Twitter API during the lockdown period to decide whether users preferred playing individually or in groups. This can help designers and developers, as well as the Saudi General Entertainment Authority (GEA), focus on creating the most interesting games for individuals and improving them to meet users' expectations. Our approach has three main stages: tweet collection, tweet preparation, and finally, the application of sentiment analysis to get the desired goal based on people's behavior toward the games. The result of this paper confirms that people, in general, preferred playing in groups, instead of alone, during this period.

**Keywords:** entertainment trends; COVID-19; Saudi Arabia lockdown; sentiment analysis; Arabic tweets





## 1. Introduction

In March 2020, the World Health Organization (WHO) [1] declared the outbreak of the novel Coronavirus disease (COVID-19), first identified in Wuhan, China, in December 2019, as a global pandemic. Since then, COVID-19 has precipitated a global crisis, with more than 214,468,601 confirmed cases and more than 4,470,969 deaths worldwide as of 27 August 2021 [1]. The progression of COVID-19 sparked a series of government interventions and responses, including country-wide lockdowns, school closures, travel bans, and cancellation of public events. Moreover, social distancing and curfews were strictly implemented to prevent the spread of the disease within communities. The COVID-19 pandemic and the need to slow its spread have impacted social interactions in nearly every sector, including employment, education, entertainment, travel, transportation, and recreation.

As millions of individuals were forced to stay home, new entertainment trends emerged [2]. Some of those trends were digital, such as video-on-demand and streaming video and electronic sell-through. Others involved physical engagement between close family members, such as playing board games and cooking new recipes. People often share their entertainment trends on social media platforms, including Twitter [3,4].

The Saudi Ministry of Health [5] confirmed the first COVID-19 case in Saudi Arabia in March 2020. The Saudi government was very proactive in preventing the spread of COVID-19. For instance, a partial curfew (6:00 AM to 7:00 PM) was issued on 23 March 2020 and lasted for 21 days. After that, a 24-h curfew was issued in major Saudi cities, including Riyadh, Dammam, Tabuk, Dahran, Hafuf, Jeddah, Taif, Qatif, and Khobar. During the Eid Al-Fitr holiday, the authorities imposed a country-wide 24-h curfew that started on 23 May 2020 and lasted for five days. Moreover, school closures were enforced between 9 March 2020 and 29 August 2021, mosque closures were enforced between 15 March 2020 and 31

May 2020, and national flight lockdowns were enforced between 19 March 2020 and 17 May 2021. Research has found an increase in gaming duration and overall online activity during these lockdown periods [6].

Sentiment analysis is a text mining technique that aims to process and detect the emotions conveyed in a given text. Typically, the goal is to help specify the attitude toward an entity, topic, or concept [7]. Here we constructed a dataset comprising Arabic tweets posted by individuals residing in Saudi Arabia using a specific set of keywords related to gaming. In this work, we study the gaming trends in Saudi Arabia during the COVID-19 lockdown (9 March 2020 to 21 June 2020) using sentiment analysis of Twitter data. The goal is to characterize the preferred gaming style, the most popular online game, and the most popular physical game among the Saudi population. We also investigate the impact of the preventative measures employed by the Saudi government on the gaming demand in Saudi Arabia. The main contributions of this work can be summarized as follows:

1. Creating a Twitter dataset containing Arabic tweets relevant to gaming trends during the lockdown period posted by individuals living in Saudi Arabia. Specific game-related keywords were used to retrieve the tweets.
2. Analyzing the sentiment of the obtained dataset to discover the Saudi community's style of gaming during the COVID-19 lockdowns (individually or in groups).
3. Analyzing the sentiment of the obtained dataset to discover the most popular online game and the most popular physical game within the Saudi community.
4. Investigating the impact of the preventative measures employed by the Saudi government on the gaming demand among the Saudi population during the lockdown period.

Our results show that 60% of people prefer to play within groups rather than play individually. We further observed that Ludo was the game most frequently played. In addition, the largest number of gaming-related tweets were posted in the week of 22 March 2020, which marks the beginning of the curfew in Saudi Arabia. The results show that people had high spirits and more interest in playing games during the curfew. The results of this work can be used as a guide by the Saudi General Entertainment Authority (GEA) as well as game designers, developers, and advertisers. Our results can assist in directing the attention of interested parties toward the gaming style, games, and gaming conditions preferred by the Saudi community and meeting their desires.

The rest of this paper is organized as follows: Section 2 discusses the related work. Section 3 presents the methodology. Sections 4 and 5 present the results and discussion, respectively. Finally, the conclusion and future works are discussed in Section 6.

## 2. Related Work

### 2.1. COVID-19 Sentiment Analysis

The COVID-19 virus encouraged researchers to work extensively on publishing research that approaches the topic from different perspectives. For example, Pokharel in [8] collected 615 tweets using Twitter API to analyze citizens' feelings during the COVID-19 outbreak in Nepal. He used the Python programming language with Tweepy and TextBlob library. The results showed that most Nepali citizens exhibited a positive reaction, while some suffered from fear and sadness. One of the biggest limitations of this study is its application to English tweets only, which does not cover the whole community since most of the tweets from this community are in the Nepali language.

Moreover, in Gupta et al. [9], a total of 12,741 tweets were collected from 5 April 2020 to 17 April 2020 with the keyword "India lockdown" to analyze feelings regarding the lockdown period using natural language processing, machine learning classifiers, and VADER lexicons. The results showed that the majority of people in India supported the government's decision to impose lockdown. However, the results of this study would be better if tweets from different phases of the lockdown were taken.

Furthermore, in order to understand and analyze the correlation between sentiment and emotions of citizens who live in neighboring countries during the COVID-19 outbreak from their tweets on Twitter, the researchers in [10] used deep Long Short-Term Mem-

ory (LSTM) models on the sentiment 140 dataset to detect users' sentiment polarity and emotions. The tweets were collected from trending hashtags in February 2020 and the publicly available Kaggle tweet dataset during the period from March and April 2020 for six neighboring countries. The results showed a high correlation in tweets' polarity for four neighboring countries, while there was opposite polarity in the other two countries. The opposite of the results that the researchers received in [11] in which citizens in the same six neighboring countries showed a similar attitude toward vaccination.

Dealing with the complexity of or the variety of dialects in the Arabic language is one of the most important issues faced by Saudi Arabian researchers. In a review of Saudi dialect in the sentiment analysis domain of Arabic tweets [12], the researchers concluded that there are still limitations and need more investigation in the tools, approaches, and techniques designed to work with Arabic. To answer the research question: How aware are Saudi citizens of the precautionary measures imposed during the COVID-19 curfew? [13], the researchers applied sentiment analysis and used an SVM classifier along with bigram in Term Frequency Inverse Document Frequency (TF-IDF) on the dataset of Arabic tweets collected during the period of the curfew. This model gave an accuracy of 85% in predicting individuals' awareness of the precautionary procedures in Saudi Arabia.

Moreover, to understand the general reaction of citizens in Saudi Arabia during COVID-19 in the Arabic domain, the researchers in [14] collected a dataset of Arabic tweets through hashtags. To achieve the desired goal, they built their lexicon and used two machine learning algorithms, NB and SVM, with 10-fold cross-validation. As a result, both classifiers gave good accuracy, but the SVM algorithm gave 98% accuracy, which is better than the NB algorithm.

Furthermore, Alsudais et al. [15] collected about 1 million Arabic tweets related to COVID-19 from the Twitter streaming API. The authors analyzed them in three different ways: identifying the topics discussed during the period, detecting rumors, and predicting the source of the tweets. They used the k-means algorithm for the first goal with k = 5. The topics discussed can be grouped as follows: COVID-19 statistics, prayers to God, COVID-19 locations, advice and education for prevention, and advertising. Authors also labeled a sample of tweets (2000 out of 1 million) annotated for false information, correct information, and unrelated, applying multiple machine learning algorithms with different sets of features. Finally, the author found that about 60% of the rumor-related tweets were classified as written by health professionals and academics, which shows the urgent need to respond to such fake news.

### 2.2. Saudi Emotions during the COVID-19 Pandemic

There are many papers analyzing the effect of the coronavirus pandemic on Saudi citizens across different sectors. For instance, in their descriptive correlational research [16], the authors aimed to examine the psychosocial effects of the COVID-19 pandemic and analyze the relationship between the use of social media and psychological stress among the population of Najran during the COVID-19 outbreak. The authors found out that the COVID-19 pandemic caused stress, anxiety, and depression, especially among the non-Saudi population of Najran in comparison to the Saudi population.

Meanwhile, Alhazmi et al. [17] tried to investigate the public's emotional response to the end of the COVID-19 lockdown in Saudi Arabia using Twitter. The authors developed an emotion detection method to classify tweets into the eight standard emotions. The finding shows that joy and anticipation are the most dominant emotions. While people expressed positive emotions, fear, anger, and sadness were also revealed. The authors of [17] show more optimistic detailed results than [16], finding that Saudi citizens were under considerably less pressure than their non-Saudi counterparts during the COVID-19 pandemic in general.

It is worth mentioning that people are usually interested in gaming and entertainment activities when they are in a good mood or have low levels of stress. Therefore, to present a staged road map of Arabic tweet sentiment analysis in favor of measuring the happi-

ness levels in Saudi Arabia cities, the authors of [18] reviewed the literature relevant to sentiment analysis, determining the sentiment of 2000 geotagged tweets in Saudi Arabia using machine learning techniques. They recorded the methodology to numerically reflect the happiness of each region, focusing more on the methodology than the results. Moreover, they mentioned that institutions of related interest, such as the GEA, could use the methodology to plan new activities and attractions.

For the same goal as [18] but using a different methodology, Alkhaldi et al. [19] aimed to analyze individuals' opinions on Twitter regarding the activities of the Saudi GEA using machine and deep learning techniques. The authors collected 3817 tweets using RapidMiner. These tweets were classified into supporters and opposers using three machine learning algorithms along with one deep learning algorithm. To evaluate the classification model, the authors used two test options: percentage split and K-fold validation tests. The results demonstrated that the people were happy and agreed with the GEA's activities. As for gender, the support rate of females was higher than males.

Based on the results of these studies, it can be posited that Saudi citizens experienced a lower level of stress during the COVID-19 pandemic [16,17], which can be a sign of their ability to entertain themselves during the curfew period.

## 3. Methodology

Our goal is to investigate the gaming styles and preferences of the Saudi population during the lockdown period. To achieve this goal, we constructed a dataset comprising Arabic tweets posted by individuals residing in Saudi Arabia using a specific set of keywords related to gaming. In this section, we discuss our dataset collection and construction, followed by the sentiment analysis process (see Figure 1).

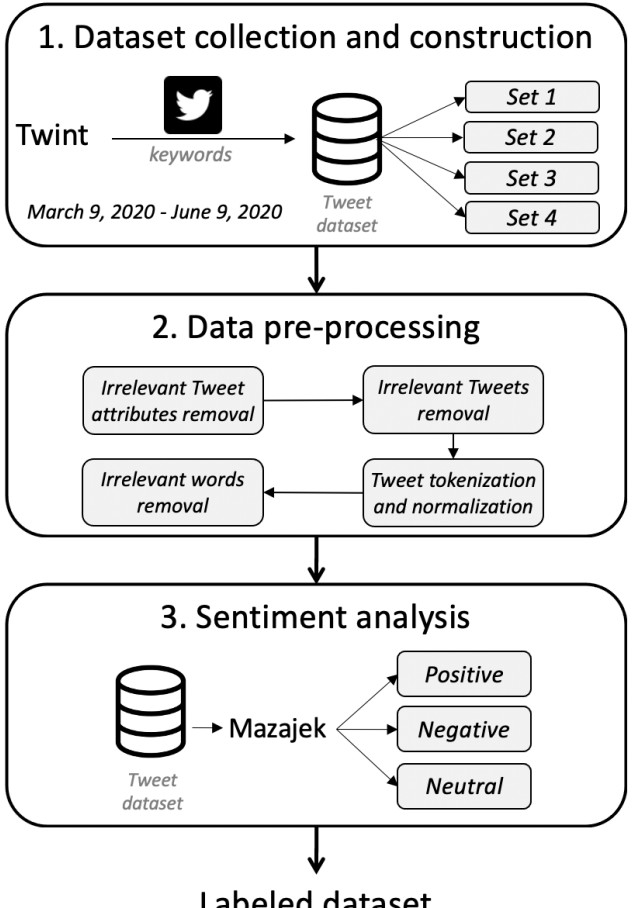

**Figure 1.** Pipeline of the methodology.

### 3.1. Dataset Collection and Construction

Our dataset is comprised of a set of Arabic tweets posted during the lockdown period by residents of Saudi Arabia (from 9 March 2020 to 21 June 2020). To collect the tweets, we used several key phrases, each indicating a different form or use of the word "play" (ألعب). The word "play" is used by Arabic speakers to express their involvement in a game (physical or virtual). Note that the root word "play" was avoided during dataset collection since it led to the retrieval of a large number of irrelevant tweets. The total number of collected tweets was 208,159. Table 1 lists the key phrases used during dataset collection. Based on the key phrases used during the tweet collection, the dataset can be partitioned into four groups, as shown in Table 1. Group one includes key phrases that indicate that a person played individually. Group two includes a single general keyword. Group three includes key phrases that suggest an invitation to play together (in the form of a question). Group four includes key phrases that state a fact related to the playing of a game. Table 1 shows the number of tweets collected in each set.

**Table 1.** Arabic key phrases used during Tweet collection and their English translation.

| Group | Key Phrase (Arabic) | Key Phrase (English) | Num of Tweets |
|:---:|:---:|:---:|:---:|
| 1 | لعبت / العب / بلعب + لحالي | I Play Alone | 1572 |
| | لعبت / العب / بلعب +لوحدي | I Play individually | 1251 |
| | لعبت / العب / بلعب + مع نفسي | I Play with Myself | 1663 |
| 2 | نلعب | We Play | 156,348 |
| | لعبنا | We Played | 19,271 |
| 3 | من+ تلعب يلعب +معي | Who Play with Me | 17 |
| 4 | تلعب / تلعبين / تلعبون + معي | You Play with Me | 5479 |
| | نلعب سوا | We Play together | 536 |

### 3.2. Dataset Preprocessing and Organization

Not all of the collected tweets in the dataset contribute to the purpose of the analysis. Therefore, the dataset was preprocessed using the following steps.

1.  Decrease the overall size of the dataset by removing irrelevant tweet attributes. The remaining attributes include the tweet text, language, mentions, URLs, photos, number of replies, number of retweets, number of likes, hashtags, quotes, and videos. These attributes were used later during the cleaning process.
2.  Clean the dataset by removing irrelevant tweets, such as tweets containing only memes, jokes, quotes, videos, and URLs. The total number of tweets after cleaning is 208,159.
3.  Clean the text of each tweet by removing English words, stop words, and vowel marks.
4.  Normalization (e.g., teh marbuta « ة » to heh « ه » and alef variants to « ا »)
5.  Tokenize each Tweet text in the dataset using the Mazajak tool [20].

### 3.3. Sentiment Analysis

We performed the sentiment analysis by assigning an emotional orientation (positive, negative, or neutral) to each tweet based on its text. A tweet is considered positive if it conveys positive opinions, feelings, or agreeable words. A tweet is considered negative if it conveys negative opinions or feelings or if it contains disagreeable or refusal words. Otherwise, a tweet can be considered neutral.

The following tweet from our dataset can be used as an example of a tweet with a positive sentiment: (وربي مبسوط مو طبيعي اول مرا افوز بقيم لعبت فيه لحالي).

The following tweet is an example of a tweet with a negative sentiment: "I played alone and got bored, so I let the enemy kill me" ( رحت لعبت لحالي و طفشت وخليت العدو يقتلني).

The following tweet is an example of a Tweet with a neutral sentiment: "I played two games yesterday alone" (لعبت قيمين البارح لحالي).

To perform our sentiment analysis, we used Mazajak [20], which is an open-source Arabic sentiment analyzer built on a convolutional neural network (CNN) followed by Long Short-Term Memory (LSTM). CNN works as a feature extractor as it can provide representative features based on local patterns that the sentences have, whereas LSTM works on these extracted features by taking context and word ordering into consideration [20].

Mazajak has been proven to achieve state-of-the-art results on many Arabic dialect datasets [20].

Figure 2 shows the sentiment of the collected tweets for each keyword; Table 2 shows the number of tweets for each keyword by grouping the keywords into four groups.

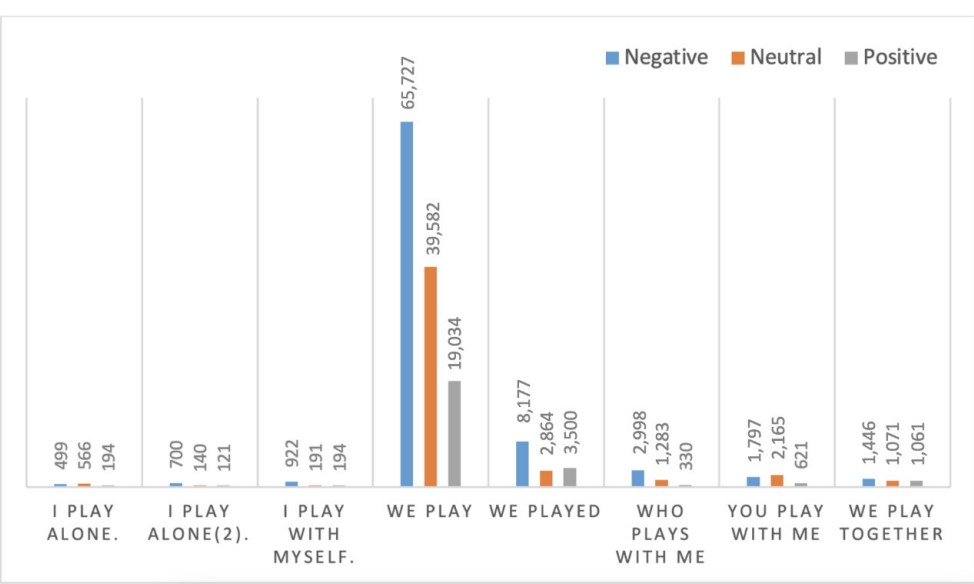

**Figure 2.** The result of sentiment analysis on collected key words.

**Table 2.** Collected data classifications results.

| Group | Sub-Group | Neutral | Positive | Negative | Total |
|---|---|---|---|---|---|
| 1 | 1.1 | 566 (44.9%) | 194 (15.4%) | 499 (39.63%) | 1259 |
| | 1.2 | 140 (14.5%) | 121 (12.5%) | 700 (72.8%) | 961 |
| | 1.3 | 191 (14.6%) | 194 (14.8%) | 922 (70.5%) | 1307 |
| 2 | 2.1 | 39,582 (27.6%) | 19,034 (13.2%) | 65,727 (45.8%) | 143,377 |
| | 2.2 | 2864 (19.6%) | 3500 (24%) | 8177 (56.23%) | 14,541 |
| 3 | 3.1 | 1283 (27.8%) | 330 (7.2%) | 2998 (65%) | 4611 |
| 4 | 4.1 | 2165 (27.2%) | 621 (13.5%) | 1797 (39.2%) | 4583 |
| | 4.2 | 1071 (29.9%) | 1061 (29.7%) | 1446 (40.4%) | 3578 |
| Total | | 47,862 (27.5%) | 25,055 (14.4%) | 82,266 (47.22%) | 174,217 |

### *3.4. Mazajak Evaluation*

This evaluation will provide us with a comprehensive understanding of the models' ability to classify tweets into positive, negative, and neutral. We fine-tuned the model on a random sample of tweets from the dataset. The sample size was about 13,000 tweets. The randomization of the sample provides a high level of representation of the whole dataset, which allows the generalization of the results.

The sample has been annotated manually into positive, negative, and neutral to evaluate the model accuracy, precision, recall, and F1 score. The results are listed in Table 3. Table 4 compares the values obtained from the Mazajak tool with those obtained from the manual annotation. Here, both neutral and positive tweets were considered positive.

**Table 3.** Mazajak results.

| F1-Score | Accuracy | Precision | Recall |
|---|---|---|---|
| 0.72 | 0.71 | 0.72 | 0.73 |

**Table 4.** Comparison between human and Mazajek results.

| True Positive | True Negative | False Positive | False Negative |
|---|---|---|---|
| 4685 | 4148 | 1865 | 1763 |

As shown in Tables 3 and 4, the Mazajak model achieved high performance (F1-score: 72% and accuracy: 71%). Further, Mazajak presents strong baselines that show a good balance between precision and recall. This indicates the power of the model in handling text sentiment analysis classification problems.

### 4. Results

In this work, we focus on analyzing three aspects of gaming during COVID-19 lockdowns: preferred gaming style (individually or within groups), preferred games (physical and online), and the impact of the preventative measures employed by the Saudi government on the gaming demand. For each type of analysis, we select a subset of tweets from our tweet dataset that contains keywords that match the analysis.

Groups one and four will be considered to determine the preferred gaming style since the other groups have unrelated and less accurate results (since group two is a general word (Games), it contains related and unrelated tweets, while group three is a question).

All four groups will be considered to gain an insight into the preferred games and the impact of preventative measures on gaming demand.

*4.1. Preferred Gaming Style*

In this analysis, we aim to identify the preferred gaming style among people living in Saudi Arabia. Specifically, we investigate if Saudis and Saudi residents preferred playing individually or within groups. Two sets from our tweet dataset, Group one and Group four were used in this analysis with a total of 11,688 tweets, as explained in Table 5.

**Table 5.** Sentiment analysis classification for the preferred gaming style.

| Group # | Group Name | Neutral | Positive | Negative | Positive + Neutral | Total |
|---------|-----------|---------|----------|----------|-------------------|-------|
| 1 | Playing Alone | 897 (26%) | 509 (14%) | 2121 (60%) | 1406 (40%) | 3527 |
| 4 | Within Groups | 3236 (40%) | 1682 (20%) | 3243 (40%) | 4918 (60%) | 8161 |
|   | Total | 4133 | 2191 | 5364 | 6324 | 11,688 |

In Table 5, it can be noticed that tweets related to playing alone are approximately half of the tweets related to playing within groups, which makes sense since people who play with others may tweet and mention each other in the same games more than in games with individual players. In addition, Table 3 shows that playing alone has 60% negative, 26% neutral, and 14% positive tweets, which can suggest that people do not like playing alone. On the contrary, there are 40% negative, 20% positive, and 40% neutral tweets about playing within groups. Therefore, results show that 60% of people prefer playing within groups over playing individually.

*4.2. Preferred Games*

In this section, we investigate the most popular game among people living in Saudi Arabia (online or physical). To do so, we searched each tweet in our dataset for game names. The search was restricted to popular games. We used the following list of popular online games: Call of duty, Pubg, Fortnite, Rainbow, and Monster; and the following list of popular physical games: Keram and Ludo. However, Ludo could be both a physical and online game.

After that, we annotate a sample tweet from each of the four groups. The manual annotation can give us more accurate sentiment analysis results.

Table 6 shows the sentiment analysis results for the selected datasets. We considered both positive and neutral as positive since the neutral context implies that they already play the game. It is important to note that Keram, which is a physical game, has the least frequent mention due to the medium that is used to discuss the games (an online platform), so the most frequently mentioned games in the collected datasets were online.

Table 6 also shows the result of the sentiment analysis of the games' keywords. We can observe that Ludo is the most frequently mentioned game, followed by Pubg, Call of duty, Fortnite, Rainbow, Keram, and finally Monster, which is the least frequently mentioned game.

**Table 6.** Games' keywords sentiment analysis.

| Keyword | In English | Positive | Negative | Neutral |
|---------|-----------|----------|----------|---------|
| كود | Call of duty | 14 | 35 | 362 |
| فورت | Fortnite | 4 | 12 | 71 |
| رينبو | Rainbow | 1 | 0 | 7 |
| كيرم | Keram | 0 | 3 | 5 |
| لودو | Ludo | 14 | 28 | 1768 |
| مونستر | Monster | 1 | 1 | 1 |

### 4.3. Impact of Preventative Measures on Gaming Demand

To analyze the impact of the preventative measures on the gaming demand, we partitioned the tweets in the dataset by their dates. Figure 3 shows the number of tweets for the period between 9 March 2020 and 9 June 2020.

From Tables 7 and 8, it can be observed that the largest number of gaming-related tweets were posted in the week of 22 March 2020. Interestingly, 22 March marks the beginning of the 7:00 p.m. to 6:00 a.m. curfew in Saudi Arabia. Moreover, 25 March was the beginning of the 3:00 p.m. to 6:00 a.m. curfew in three major cities (Riyadh, Makkah, and Almadinah).

**Table 7.** Highest game-related tweeting per day.

| date | 26/03 | 25/03 | 24/03 | 27/03 | 23/03 | 28/03 | 22/03 |
|---|---|---|---|---|---|---|---|
| tweets | 3008 | 2871 | 2833 | 2817 | 2799 | 2767 | 2646 |

**Table 8.** Lowest game-related tweeting per day.

| date | 21/06 | 09/03 | 24/04 | 25/04 | 10/03 | 29/04 | 26/04 |
|---|---|---|---|---|---|---|---|
| tweets | 3008 | 2871 | 2833 | 2817 | 2799 | 2767 | 2646 |

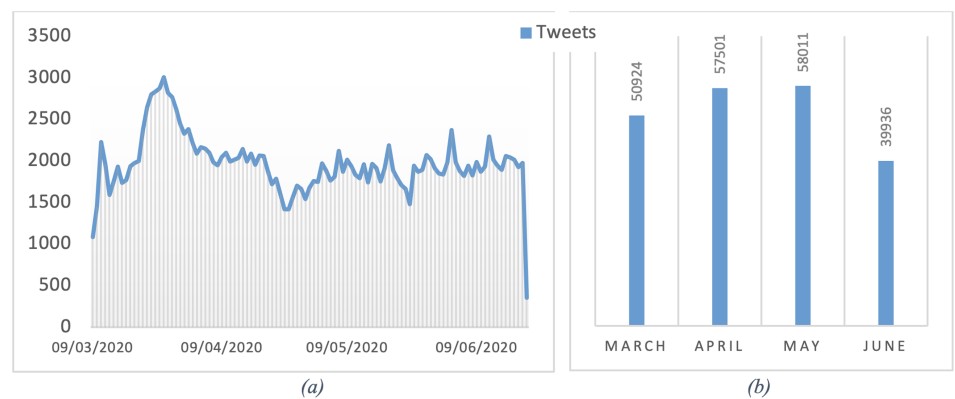

**Figure 3.** Game-related tweets. (**a**) Tweeting trends sorted by day. (**b**) Tweeting trends sorted by month.

## 5. Discussion

This study investigates the gaming styles and preferences of Saudi users. It also analyzes the impact of the different preventive measures employed by the Saudi government on the gaming demand among Saudi users.

The two royal orders of partial and full curfews issued in the same week affected the gaming demand. Thus, it can be noted that these results agree with [17], whose finding shows that joy and anticipation are the most dominant emotions in Saudi. Moreover, our results may also agree with the authors in [16], who found that Saudi citizens experienced a lower level of pressure than non-Saudis during the COVID-19 pandemic in general. Interestingly, the lowest tweeting rate per day was observed on the day the curfew lifted.

The current study has some limitations. First, confining Arabic keywords during tweet collection was challenging due to the presence of different dialects. For example, the four words (بلحالي، لوحدي، لحالي، مع نفسي) all translate to "alone".

Second, some of the collected tweets in the dataset were irrelevant to the sentiment analysis task after preprocessing. This includes tweets in which people discuss their memories of games.

Finally, searching tweets based on location imposed an extra challenge since many Twitter users disable their geographic location. Hence, the targeted audience is inaccurate because there is a high possibility that we collected tweets from other countries in addition to Saudi tweets.

## 6. Conclusions

To analyze people's behavior during the COVID-19 lockdown period, we collected a dataset of game-related keywords using Twitter API to see whether Saudi people preferred to play alone or within groups. The sentiment analysis was performed using the Mazajak sentiment analyzer. A sample with 12,462 tweets was randomly selected from our collected dataset to evaluate the Mazajak model; the evaluation shows that Mazajak gives good results on Arabic tweets. The result of this analysis confirmed that people, in general, preferred playing in groups over playing alone during this period. This can help designers, developers, and the Saudi GEA to focus on meeting people's desires. In future work, we plan to obtain more specific results by focusing on the names of the games when comparing people's preferences to decide whether people preferred online or physical games during the lockdown period, the entertainment direction in general, and how they spent their time during the lockdown period. Furthermore, we aim to compare different Arabic language sentiment analyzers and use araBERT, which has become available online recently (5 months ago) and has a pre-trained model for sentiment analysis tasks in Arabic.

**Author Contributions:** Conceptualization, A.A. and R.A.; writing—original draft preparation, A.A. and R.A.; writing—review and editing, A.A., R.A. and H.A.; supervision, H.A. All authors have read and agreed to the published version of the manuscript.

**Funding:** This research received no external funding.

**Institutional Review Board Statement:** Not applicable.

**Informed Consent Statement:** Not applicable.

**Data Availability Statement:** The dataset and analysis codes are available at https://github.com/fdAmaal/SaudiCurefewGames (accessed on 19 April 2022).

**Conflicts of Interest:** The authors declare no conflict of interest.

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
