# Peer review of "Understanding Entertainment Trends during COVID-19 in Saudi Arabia"

_information, doi:10.3390/info13070308_

Round 1
Reviewer 1 Report
The article is on an interesting topic and is easy to read and follow as well. The methodology seems good and the authors have obtained interesting findings.
The related work section however is limited. Since the covid outbreak, there are a plethora of good-quality papers which are missing here.
For instance, the authors cited [8] and [9], but forgot to refer to the highly cited article on the same topic in related work under section 2.1.
i. cross-cultural polarity and emotion detection using sentiment analysis and deep learning on covid-19 related tweets
https://ieeexplore.ieee.org/abstract/document/9207881
ii. Evaluating polarity trend amidst the coronavirus crisis in peoples' attitudes toward the vaccination drive
https://www.mdpi.com/2071-1050/13/10/5344
The authors mentioned they used Mazejek for sentiment analysis. I wonder if there are other open-source Arabic language sentiment analyzers available in the market or not. If so, I suggest authors compare results using other algorithms. Also, is Mazajek lexicon-based?
Were the tweets collected in Arabic or they were translated from English to Arabic?
Nowadays, embeddings-based deep learning architectures have proven significantly well. Have auhors looked into pre-trained embedding models for sentiment analysis task such as BERT for Arabic language?
- Missing ref. /additional brackets on line 16.
Author Response
Response to Reviewer 1 Comments
We thank the reviewer for their reading of the manuscript and for their constructive comments. We have taken the comments into consideration to improve the quality of the manuscript. Please find below a point by point response to each comment.
Point 1.1: The related work section however is limited. Since the covid outbreak, there are a plethora of good-quality papers which are missing here. For instance, the authors cited [8] and [9], but forgot to refer to the highly cited article on the same topic in related work under section 2.1.
- cross-cultural polarity and emotion detection using sentiment analysis and deep learning on covid-19 related tweets
https://ieeexplore.ieee.org/abstract/document/9207881
- Evaluating polarity trend amidst the coronavirus crisis in peoples' attitudes toward the vaccination drive
https://www.mdpi.com/2071-1050/13/10/5344
Response 1.1: The first suggested reference has been added to the manuscript (reference 10 in the revised manuscript)and has been cited in the related work section (Section 2.1 - line: 91). Specifically, we added the following sentence to express this reference:
“Furthermore, In order to understand and analyze the correlation between sentiment and emotions of citizens who live in neighboring countries during Covid-19 outbreak from their tweets on Twitter, the researcher in [10] used deep long short-term memory (LSTM) models on the sentiment 140 data set to detect users' sentiment polarity and emotions. The tweets were collected from trending hashtags in February 2020, and publicly available Kaggle tweet data set during the period from March and April 2020 for 6 neighboring countries. The results showed a high correlation in tweets’ polarity for four neighboring countries, while there was opposite polarity in the other two countries.”
The second suggested reference has also been added to the manuscript (reference 11 in the revised manuscript) and has been cited in the related work section (Section 2.1 - line: 98). Specifically, we added the following sentence to express this reference:
“As opposite of the results that researchers get in [11], in which citizens in the same six neighboring countries showed a similar attitude toward the vaccination.”
Point 1.2: The authors mentioned they used Mazejek for sentiment analysis. I wonder if there are other open-source Arabic language sentiment analyzers available in the market or not. If so, I suggest authors compare results using other algorithms.
Response 1.2: Yes, other open-source Arabic language sentiment analysers are available. However, Mazajak is considered a well-known Arabic language sentiment analyser since it already has been evaluated and proved to provide high results. In Section 3.3 (Sentiment Analysis) - lines: 222 - 229 we mentioned a few references that reported the high performance that Mazajak achieved.
In addition, comparing different Arabic language sentiment analysers seems to be a very interesting contribution that we plan to keep for another manuscript. We added this as a future work in the Conclusion Section (lines 314-315).
Point 1.3: Is Mazajek lexicon-based?
Response 1.3: The creators of Mazajak do not specifically describe the tool as lexicon-based. Accordingly, we did not report it as such in our manuscript.
Point 1.4: Were the tweets collected in Arabic or they were translated from English to Arabic?
Response 1.4: Collected tweets were all written originally in Arabic. We added this piece of information to the manuscript in Section 3.1 (Dataset Collection and Construction) - line 171.
Point 1.5: Nowadays, embeddings-based deep learning architectures have proven significantly well. Have auhors looked into pre-trained embedding models for sentiment analysis task such as BERT for Arabic language?
Response 1.5: Mazajak is considered an embeddings-based deep learning tool. It achieves good results on many Arabic dialects; therefore, we used it in our manuscript.
araBERT, is another pre-trained model available for sentiment analysis tasks of Arabic texts. However, because it was released very recently (about 5 months ago), we are considering this model for a future work. We added a comment about this in the Conclusion Section (lines 315-316).
Point 1.6: Missing ref. /additional brackets on line 16.
Response 1.6: Resolved.

Reviewer 2 Report
The paper focuses on understanding entertainment trends during the COVID-19 pandemic.
The topic is interesting and worth investigating. The paper includes a comprehensive enough literature review. However several issues should be addressed, as mentioned in the following.
Recommendations:
- The main issue of the paper is that the authors do not evaluate how well the classification using Mazajak works (in terms of Precision, Recall, Accuracy, F1-Score). Without such an evaluation the results included in the paper cannot be considered reliable.
- The literature review could be extended to also mention works focusing one covid-19 vaccine stance analysis
Author Response
Response to Reviewer 2 Comments
We thank the reviewer for their reading of the manuscript and for their constructive comments. We have taken the comments into consideration to improve the quality of the manuscript. Please find below a point by point response to each comment.
Point 2.1: The main issue of the paper is that the authors do not evaluate how well the classification using Mazajak works (in terms of Precision, Recall, Accuracy, F1-Score). Without such an evaluation the results included in the paper cannot be considered reliable.
Response 2.1: The model was not evaluated in the manuscript because training it was not part of our work. In our case, we used a well-known pre-trained model which has been already evaluated and which achieves state-of-the-art results on many Arabic dialect datasets. We added a comment about this to the manuscript in lines 228-229 - Section 3.3 (Sentiment analysis).
Point 2.2: The literature review could be extended to also mention works focusing one covid-19 vaccine stance analysis
Response 2.2: The Tweet dataset was collected before any COVID-19 vaccine became available (the period from March 9, 2020 to June 21, 2020).

Round 2
Reviewer 2 Report
I would like to thank the authors for their answer and the changes made. However I do not agree with their justifications:
- Response 2.1: The fact that the model has been evaluated and worked well on other datasets does not necessarily imply that it also works well on the dataset used in this paper. I don't think that the paper can be considered scientifically sound without a proper evaluation in terms of Precision, Recall, Accuracy, F1-Score.
- Response 2.2: The paper has been submitted after COVID-19 vaccines have become available. However, mentioning this was just a suggestion and I perfectly agree with the authors if they choose not to mention this.
The revised version includes missing references "results that researchers get in [? ],". The actual reference should be mentioned instead of "?".
Author Response
We have taken the comments into consideration to improve the quality of the manuscript.
Please find below a point by point response to each comment.
Point 2.1: The fact that the model has been evaluated and worked well on other datasets does
not necessarily imply that it also works well on the dataset used in this paper. I don't think that
the paper can be considered scientifically sound without a proper evaluation in terms of
Precision, Recall, Accuracy, F1-Score.
Response 2.1: As suggested by the reviewer, the Mazajek tool was evaluated in Section 3.4 in
the revised manuscript.
Point 2.2: The paper has been submitted after COVID-19 vaccines have become available.
However, mentioning this was just a suggestion and I perfectly agree with the authors if they
choose not to mention this.
Response 2.2: We will consider the reviewer’s suggestion for future research.
Point 2.3: The revised version includes missing references "results that researchers get in [?
],". The actual reference should be mentioned instead of "?".
Response 2.3: The missing reference has been added in the revised manuscript (reference
number 20). The citation was mention in line 99.

Round 3
Reviewer 2 Report
The authors are encouraged to also add a few comments in the paper concerning the values of the metrics computed in Section 3.4.
Author Response
Response to Reviewer 2 Comments
We thank the reviewer for their reading of the manuscript and for their constructive comments. We have taken the comments into consideration to improve the quality of the manuscript. Please find below a point by point response to each comment.
Point 2.1: The authors are encouraged to also add a few comments in the paper concerning the values of the metrics computed in Section 3.4.
Response 2.1: As suggested by the reviewer, a paragraph that discusses the results of the metrics presented in Table 3 was added to the manuscript.